# Evolutionary history and classification of Micropia retroelements in Drosophilidae species

Juliana Cordeiro[1]*, Tuane Letícia Carvalho[2], Vera Lúcia da Silva Valente[3], Lizandra Jaqueline Robe[2,4]

**1** Departamento de Ecologia, Zoologia e Genética, Instituto de Biologia, Universidade Federal de Pelotas, Pelotas, RS, Brazil, **2** Programa de Pós-Graduação em Biodiversidade Animal, Universidade Federal de Santa Maria, Santa Maria, RS, Brazil, **3** Departamento de Genética, Instituto de Biociências, Universidade Federal do Rio Grande do Sul, Porto Alegre; Rio Grande do Sul; Brazil, **4** Departamento de Ecologia e Evolução, Centro de Ciências Naturais e Exatas, Universidade Federal de Santa Maria, Santa Maria, RS, Brazil

☯ These authors contributed equally to this work.

\* jlncdr@gmail.com

**Data Availability Statement:** All relevant data are within the paper and its Supporting Information files.

## Abstract

Transposable elements (TEs) have the main role in shaping the evolution of genomes and host species, contributing to the creation of new genes and promoting rearrangements frequently associated with new regulatory networks. Support for these hypotheses frequently results from studies with model species, and *Drosophila* provides a great model organism to the study of TEs. Micropia belongs to the Ty3/Gypsy group of long terminal repeats (LTR) retroelements and comprises one of the least studied *Drosophila* transposable elements. In this study, we assessed the evolutionary history of Micropia within Drosophilidae, while trying to assist in the classification of this TE. At first, we performed searches of Micropia presence in the genome of natural populations from several species. Then, based on searches within online genomic databases, we retrieved Micropia-like sequences from the genomes of distinct Drosophilidae species. We expanded the knowledge of Micropia distribution within *Drosophila* species. The Micropia retroelements we detected consist of an array of divergent sequences, which we subdivided into 20 subfamilies. Even so, a patchy distribution of Micropia sequences within the Drosophilidae phylogeny could be identified, with incongruences between the species phylogeny and the Micropia phylogeny. Comparing the pairwise synonymous distance (dS) values between Micropia and three host nuclear sequences, we found several cases of unexpectedly high levels of similarity between Micropia sequences in divergent species. All these findings provide a hypothesis to the evolution of Micropia within Drosophilidae, which include several events of vertical and horizontal transposon transmission, associated with ancestral polymorphisms and recurrent Micropia sequences diversification.

**Funding:** Funding was provided by the Conselho Nacional de Desenvolvimento Científico e Tecnológico (CNPq - Universal 402447/2016-6), Coordenação de Aperfeiçoamento de Pessoal de Nível Superior (CAPES), and the Fundação de Amparo à Pesquisa do Estado do Rio Grande do Sul (FAPERGS). The funders had no role in study design, data collection and analysis, decision to publish, or preparation of the manuscript.

**Competing interests:** The authors have declared that no competing interests exist.

## Introduction

Since Barbara McClintock's first publication on maize transposable elements (TEs), these sequences went from junk to pivotal characters in the control and evolution of genomes. The discovery of unexpected high amounts of TEs in the genome of distinct species has pointed out toward functions of TEs on these genomes [1, 2, 3]. In fact, current knowledge indicates that TEs have been shaping the evolution of genomes and host species [4], contributing to the creation of new genes [5, 6] and promoting rearrangements frequently associated with new regulatory networks [7, 8, 9]. Moreover, there is evidence that TEs may assist in the control of embryonic development [9, 10] and genomic plasticity [11].

A large fraction of most eukaryotes genome is composed of TEs known as retroelements [12, 13, 14], some of which belong to the long terminal repeats (LTR) order. Phylogenetic analyses of such retroelements reveal an evolutionary history consisting mainly of vertical transposon transmissions (VTT) and intraspecific diversification [15]. However, autonomous TEs are able to invade naïve genomes through horizontal transposon transfers (HTT), in which they make copies of themselves and evade host defense systems before becoming fully silenced by genomic anti-TE mechanisms [16, 17]. Although HTTs are still considered rare events, mainly because we can only detect the successful ones, it seems that such events represent an important step in the TEs' life cycle. This step enables them to evade the natural progression of their birth-and-death process [18, 19, 17, 16]. After the HTT event, TEs can have a wide range of positive and/or negative consequences in the host genome [20]; but mainly, they become a new set of sequences where evolution can take place, unveiling their relevance to host genome evolution [21, 22].

A growing number of studies have identified HTTs using distinct analysis strategies [15, 16, 23, 24, 25]. For instance, a patchy taxonomic distribution among monophyletic clusters of species is expected if TEs are moving horizontally rather than being vertically inherited. This patchy distribution associated with incongruences between species and TEs phylogenies, as well as an unexpected high nucleotide identity between TEs found in the genome of divergent species, widely strengthens the evidence for HTT [26, 17, 25, 27, 28]. According to these criteria, LTR retrotransposons account for approximately 20% of HTT events across the genomes of insect [16]. This value increases when only *Drosophila* genomes are analyzed, e.g. LTR retroelements account for 90% of the HTT events detected across the genomes of *D. melanogaster*, *D. simulans* and *D. yakuba* [29].

Micropia is a retrotransposon that belongs to the Ty3/Gypsy group of LTR retroelements [30], which is closely related to retroviruses [31, 32]. Micropia was first discovered in the lampbrush loops of the *Drosophila hydei* Y chromosomes. Until recently, there were only four well-characterized Micropia elements, and these were found in the genomes of *D. hydei* (named dhMiF2 and dhMiF8) and *D. melanogaster* (named Dm11 and Dm2) [33, 34, 35]. Recently, complete and probably active Micropia reference sequences were found in the genomes of *D. simulans* and *D. sechellia* [15]. Nevertheless, Micropia related sequences are also present in the genomes of several *Drosophila* and *Zaprionus* species, showing an irregular distribution pattern [36, 37, 38, 39, 40, 41]. In some species (like *D. hydei*), Micropia shows an effective transcription-based repression mechanism associated with antisense RNAs [37, 41, 42]. But there is no evidence of autonomous Micropia sequences for other species, for example *D. melanogaster* [41].

Here, our goals were to provide a hypothesis to the evolutionary history of Micropia retroelement sequences within Drosophilidae species genome, while trying to assist in the classification of this TE. At first, we analyzed its presence in the genome of natural populations from several species and sequenced the detected elements. Then, we expanded our data set based on

searches for Micropia-like sequences within genomic databases. All these sequences were used to propose a hypothesis to the evolution of Micropia within Drosophilidae while assessing its subdivision and identifying several cases of HTTs.

## Materials and methods

### Species analyzed

For this study, we analyzed the presence/absence of Micropia sequences in the genomes of natural populations of 24 *Drosophila* species. These species were field-collected during 2000–2009 or obtained at the Tucson *Drosophila* Stock Center (current National *Drosophila* Species Stock Center at Cornell University) (Table 1). To this end, PCR-blot and Dot-blot searches (hereafter "*in vitro searches*") were performed following the methodology described in *In vitro searches*: *DNA manipulation*, *PCR-blot*, *Dot-blot*, *and sequencing* (see below). *In vitro* searches were also previously performed for the other three species of the *cardini* group [39], and for *D. melanogaster* [34, 35] and *D. hydei* [33]; the sequences thus obtained were downloaded from Gen-Bank. We also analyzed the presence/absence of Micropia sequences in 26 species, whose genomes are available at NCBI (blast.ncbi.nlm.nih.gov/Blast.cgi) or Flybase (flybase.org/blast/) websites (hereafter "*in silico* searches"), plus two species, *D. suzukii* and *D. buzzatii*, whose genomes are available at personal websites (http://spottedwingflybase.org/ and https://dbuz.uab.cat/welcome.php, respectively) (Table 1). These searches followed the criteria described in *In silico searches*: *Genomic analysis* (see below). Thus, *D. buzzatii*, and *D. melanogaster* were the only species for which both search strategies were applied. The classification scheme adopted for each of these species across this study follows the proposal of [43].

### *In vitro* searches: DNA manipulation, PCR-blot, dot-blot, and sequencing

Genomic DNA was extracted through phenol-isoamyl-chloroform protocol according to [44]. It was used approximately 100 adult flies *per* species macerated in liquid nitrogen using individual new sterile grinders. PCR reactions were performed using Micropia primers to amplify the reverse transcriptase (RT) domain within the *pol* gene, as described in [39]. The following conditions were used for 25 µl PCR reactions: 25 ng of template DNA, 20 pmol of each primer, 0.2 mM of each nucleotide, 1.5 mM MgCl2 and 1 unit Taq DNA polymerase in 1x polymerase buffer (all from Invitrogen). Amplifications parameters were 95˚C for 2 min, 35 cycles at 95˚C for 30 s, 50–60˚C for 30 s and 72˚C for 1 min, followed by an extension step at 72˚C for 10 min. *Drosophila hydei* genomic DNA was used as a positive control.

In order to confirm the homology of the amplified fragments, a PCR-blot was prepared with the obtained PCR amplicons. The PCR products were separated by electrophoresis using a 1% agarose gel and transferred to nylon membranes (Hybond N+®, GE Healthcare), where hybridization was carried out using an 812 bp fragment of Micropia from *D. hydei* as the probe. This fragment ranges from nucleotide 1,777 to 2,589 of the *D. hydei* dhMiF2 sequence (GenBank acc. no. X133041), covering part of the RT sequence. The probe label and signal detection were performed using the Gene ImagesTM AlkPhos DirectTM labeling and detection system (GE Healthcare), according to the manufacturer's instructions. The membranes were hybridized at 55˚C and exposed for 5 min.

A Dot-blot procedure was also performed using genomic DNA. Denaturation was performed using 3 µg of genomic DNA in a final volume of 10 µl, which was directly applied onto a nylon membrane (Hybond N+®, GE Healthcare). As a positive control, 5 ng (in 10 µl) of the dhMiF2 probe was used. The probe labeling, signal detection, and hybridization temperature were performed as above. The Dot-blot revealing film underwent 3 min exposure.

**Table 1. Presence/absence of Micropia sequences in the genomes of Drosophilidae species.** Methodology employed and GeneBank accession numbers are also shown.

| Genus | Subgenus | Group species | Species | Presence/absence | Methodology | GenBank acc. nos. |
|---|---|---|---|---|---|---|
| *Drosophila* | *Dorsilopha* | *busckii* | *D. busckii* | + | *in silico* | see S1 Table |
| *Drosophila* | *Drosophila* | *cardini* | *D. acutilabella*[A] | + | *in vitro* | FJ748684*, FJ748685*, FJ748686*, FJ748687*, FJ748688* |
| | | | *D. arawakana*[E] | - | *in vitro* | - |
| | | | *D. cardini*[E] | + | *in vitro* | FJ748690*, FJ748691*, FJ748692* |
| | | | *D. cardinoides*[E] | + | *in vitro* | EF090263*, EU149929*, EU149930* |
| | | | *D. dunni*[A] | - | *in vitro* | - |
| | | | *D. neocardini*[E] | + | *in vitro* | EF090264*, EU149931*, EU149932*, EU149933* |
| | | | *D. neomorpha*[A] | + | *in vitro* | FJ748695*, FJ748696*, FJ748697* |
| | | | *D. nigrodunni*[A] | - | *in vitro* | - |
| | | | *D. parthenogenetica*[A] | + | *in vitro* | FJ748698*, FJ748699*, GQ339587*, GQ339588*, GQ339589*, GQ339590* |
| | | | *D. polymorpha*[E] | + | *in vitro* | EF090265*, EF149934*, EF149935*, EF149936*, EF149937* |
| | | | *D. procardinoides*[A] | + | *in vitro* | FJ748700*, FJ748701*, FJ748702* |
| | | | *D. similis*[A] | - | *in vitro* | - |
| | | *funnebris* | *D. funnebris*[A] | - | *in vitro* | - |
| | | *guaramunu* | *D. griseolineata*[D] | - | *in vitro* | - |
| | | | *D. maculifrons*[D] | - | *in vitro* | - |
| | | *guarani* | *D. guaru*[D] | - | *in vitro* | - |
| | | | *D. ornatifons*[D] | - | *in vitro* | - |
| | | *immigrans* | *D. albomicans* | + | *in silico* | see S1 Table |
| | | | *D. immigrans*[D] | - | *in vitro* | - |
| | | *tripunctata* | *D. bandeirantorum*[B] | - | *in vitro* | - |
| | | | *D. mediodiffusa*[B] | - | *in vitro* | - |
| | | | *D. mediopictoides*[B] | - | *in vitro* | - |
| | | | *D. mediopunctata*[B] | - | *in vitro* | - |
| | | | *D. paraguayensis*[C] | - | *in vitro* | - |
| | | | *D. paramediostriata*[B] | - | *in vitro* | - |
| | | | *D. tripunctata*[B] | - | *in vitro* | - |
| | *Siphlodora* | *repleta* | *D. arizonae* | + | *in silico* | see S1 Table |
| | | | *D. buzzatii*[C] | + | *in vitro/ in silico* | FJ748689*, GQ339579*, GQ339580*, GQ339582*, see S1 Table |
| | | | *D. hydei*[C] | + | *in vitro* | X13304*, X13305* |
| | | | *D. mercatorum*[C] | + | *in vitro* | FJ748693*, FJ748694*, GQ339583*, GQ339584*, GQ339585* GQ339586* |
| | | | *D. mojavensis* | + | *in silico* | see S1 Table |
| | | | *D. navojoa* | + | *in silico* | see S1 Table |
| | | | *D. zottii* | + | *in vitro* | FJ748703*, GQ339578* |
| | | *virilis* | *D. americana* | + | *in silico* | see S1 Table |
| | | | *D. virilis* | + | *in silico* | see S1 Table |
| | *Sophophora* | *melanogaster* | *D. ananassae* | + | *in silico* | see S1 Table |
| | | | *D. bipectinata* | + | *in silico* | see S1 Table |
| | | | *D. elegans* | + | *in silico* | see S1 Table |
| | | | *D. erecta* | + | *in silico* | see S1 Table |
| | | | *D. ficusphila* | + | *in silico* | see S1 Table |
| | | | *D. kikkawai* | + | *in silico* | see S1 Table |
| | | | *D. melanogaster*[E] | + | *in vitro/in silico* | X14037*, X14173*, see S1 Table |

*(Continued)*

**Table 1.** (Continued)

| Genus | Subgenus | Group species | Species | Presence/absence | Methodology | GenBank acc. nos. |
|---|---|---|---|---|---|---|
| | | | *D. rhopaloa* | + | *in silico* | see S1 Table |
| | | | *D. sechellia* | + | *in silico* | see S1 Table |
| | | | *D. simulans* | + | *in silico* | see S1 Table |
| | | | *D. suzukii* | + | *in silico* | see S1 Table |
| | | | *D. takahashii* | + | *in silico* | see S1 Table |
| | | | *D. yakuba* | + | *in silico* | see S1 Table |
| | | obscura | *D. Miranda* | - | *in silico* | - |
| | | | *D. persimilis* | - | *in silico* | - |
| | | | *D. subobscura* | - | *in silico* | - |
| | | willistoni | *D. willistoni* | + | *in silico* | see S1 Table |
| | *Haiwaiian Drosophila* | - | *D. grimshawi* | - | *in silico* | - |
| *Phortica* | - | *variegata* | *P. variegata* | - | *in silico* | - |
| *Scaptodrosophila* | - | - | *S. lebanonensis* | + | *in silico* | see S1 Table |

*Sequences used as initial BLASTn queries. Capital letters refer to the fly collector/supplier:

[A]Tucson *Drosophila* Stock Center (currently The National *Drosophila* Species Stock Center at Cornell University)

[B]Dr. Luciano Basso da Silva

[C]Dr. Marco Silva Gottschalk

[D]MSc. Jonas da Silva Doge

[E]Dra. Daniela Cristina De Toni. Species vouchers are available at the Laboratório de Drosophilidae at Universidade Federal do Rio Grande do Sul.

For sequencing, PCR amplicons from each species presenting positive signals for Micropia were separated by 1.5% agarose gel electrophoresis and purified using Illustra GFXTM PCR DNA and Gel Band Purification kit (GE Healthcare) according to the supplier's specifications. The fragments were cloned using pGEM®-T Easy Vector system (Promega). The obtained recombinant plasmids underwent a new PCR reaction using the universal M13 primers at a 55˚C annealing temperature. The amplicons were purified using ExoI-SAP (GE Healthcare) and directly sequenced in a MegaBACETM500 (GE Healthcare). Forward and reverse strands were sequenced; ambiguities and compressions were resolved through assemblage in the Staden Package Gap4 program [45]. GenBank accession numbers are indicated in Table 1.

### *In silico* searches: Genomic analysis

BLAST searches were performed at NCBI (blast.ncbi.nlm.nih.gov/Blast.cgi) and Flybase website (flybase.org/blast/), using default parameters against "Whole Genome Shotgun Contigs (WGS) database" limited by "organism", in which each *Drosophila* species was selected. For *D. buzzatii* and *D. suzukii*, searches were performed against the scaffolds database, respectively, in the '*Drosophila buzzatii* Genome Project' website (dbuz.uab.cat/welcome.php) and in the 'Spotted Wing FlyBase' website (spottedwingflybase.org/). The searches were finished in January 2018.

The initial BLASTn queries consisted of Micropia reverse transcriptase (RT) nucleotide sequences obtained by previous studies [39, 33, 34 and 35] and retrieved from GeneBank (Table 1). The retrieved sequences obtained during the *in silico* searches showing scores higher than 50 and E-values lower than 1.0E-05 were downloaded, including 2 kb from both sides of each hit. After that, each retrieved sequence was aligned with the set of query sequences using ClustalW, as implemented in MEGA6 software [46]. Sequences that failed to align in this first step of multiple alignment underwent a second step of alignment (this time pairwise or even

local alignment) against the query sequence which presented the highest score in the BLASTn searches (hereafter "best query" sequence). In this case, fragments presenting less than 300 bp of confirmed homology to its best query sequence were withdrawn from the alignment. Furthermore, after compressing the analyzed region, identical nucleotide sequences recorded for the same species were joined in a single sequence.

A codon-based alignment was then performed using Muscle [47] as implemented in MEGA6 software. Gaps presented in this matrix were further resolved, in order to leave all sequences in-frame, to obtain the aligned amino acid matrix. All these translated sequences were then used as queries to perform exhaustive tBLASTn searches, using the same strategy described above. These two-BLAST-step strategy was performed to guarantee that the real diversity of Micropia sequences was retrieved from the genomes, enabling a better representation of these sequences in our data set. Supplementary S1 Table provides a list of BLASTn and tBLASTn results, whereas S1 File provides the set of nucleotide sequences retrieved through "*in vitro*" and "*in silico*" searches. The first analyzed matrix encompassed all sequences obtained under these criteria that presented a minimum overlap of 300bp to the previous nucleotide alignment, after a final codon-based alignment performed in Muscle (first filtering step, resulting in S2 and S3 Files).

After completing the matrix, putative functional RT Micropia sequences were identified by translating each unaligned nucleotide sequence in the different reading frames. Once an Open Reading Frame (ORF) was detected, BLASTn searches further confirmed its identity.

## Phylogenetic analysis and Micropia subfamilies

Phylogenetic analyses were performed using the amino acid alignment obtained after resolving all gaps and leaving all nucleotide sequences in-frame. Fifty amino acid sequences belonging to each of the five main clades recently established by Bargues and Lerat [15] for the Micropia/Sacco group within Ty3/Gypsy were selected from the alignment provided by the authors. These sequences were included as a "taxonomic framework" to guide conclusions related to new Micropia sequences in our phylogenetic analyses, in which a Copia-like transposable element sequence obtained from the *D. melanogaster* genome (GenBank access number X01472) was used as outgroup. This Copia-like retroelement belongs to the Ty1/Copia superfamily of LTR retrotransposons, which is closely allied to the Ty3/Gypsy retrotransposon sequence group [48].

Bayesian phylogenetic analysis (BA) was performed under a mixed model with gamma correction, as implemented in MrBayes3.1.2 software, through Cipres Computational Resources [49]. This Markov Chain Monte Carlo (MCMC) search was run for 10,000,000 generations, with trees saved every 1,000 after a burn-in of 2,500. The Posterior Probability (PP) of each clade on the 50% majority-rule consensus tree was calculated and the resulting tree was visualized in FigTree. The tree so obtained was used to detect intraspecific sequences sharing a most recent common ancestor (MRCA). In these cases, only the sequence with the shortest branch (the most similar to the inferred MRCA sequence) was maintained as representative of that clade in a new round of BA analysis (second filtering step, resulting in S4 File). The final tree was compared to the species tree, as compiled from previous studies [50, 51, 52, 53, 54, 55 and 56], which present only a limited overlap on sampled species. Subfamilies of the Micropia TE sequences were identified using the criterion established by Capy et al. [30], according to which reciprocally monophyletic sequences with less than 30% of divergence at the amino acid level could be grouped in the same TE subfamily. This analysis was performed in MEGA6, using Poisson amino acid substitution model.

### dS and divergence time estimates

Pairwise synonymous distance (dS) values were estimated for Micropia in-frame nucleotide sequences (S5 File) and for three host nuclear genes sequences (S2 Table) using Nei and Gojo-bori [57] method, as implemented in MEGA6. Alcohol-dehydrogenase (*Adh*), alpha–methyl-dopa (*Amd*) and dopa-decarboxylase (*Ddc*) sequences were downloaded from GeneBank or retrieved from the species genomes using BLASTn searches (for GenBank or scaffold accession numbers, see S2 Table). In order to identify if the Micropia dS values were significantly lower than those observed for the host nuclear genes, accounting for differences in the number of synonymous sites, a one-tailed Fisher's exact test was performed using R v.3.5.2 [58]. Divergence times were also eventually evaluated using dS estimates and a synonymous substitution rate of 0.016 substitutions per site per million years, as calculated for *Drosophila* genes with low codon usage bias [59].

## Results

### Species analyzed

A total of 56 Drosophilidae species were analyzed for the presence/absence of Micropia sequences (Table 1). Thirty species were analyzed by *in vitro* searches and 28 species were analyzed through *in silico* searches. *In vitro* and *in silico* searches allowed to isolate 363 Micropia sequences plus one outgroup sequence (S3 Table and S1 File), which were further reduced to 247 plus one outgroup sequence (S2 and S3 Files) in the first filtering step. The second filtering step followed by the inclusion of the Micropia/Sacco sequences characterized by [15], leads to the alignment of 151 sequences (S4 File).

### Patchy distribution of Micropia sequences in the Drosophilidae species genomes

We identified the presence of distinct Micropia related sequences in the genome of 34 Drosophilidae species (Table 1). *In vitro* signals of Micropia copies were encountered in *D. melanogaster* and in some species from *cardini* (8 of the 12 species tested) and *repleta* (4 of the 4 species tested) groups, despite the fact that 13 other species were also tested (Table 1, S1 and S2 Figs). Conversely, *in silico* searches enabled the isolation of Micropia sequences in the genomes of *D. buscki*, *D. albomicans*, *D. willistoni* and *S. lebanonensis*, and in species from the *repleta* (4 of the 4 species tested), *virilis* (2 of the 2 species tested) and *melanogaster* (12 of the 12 species tested) groups. No Micropia sequence could be found for *D. grimshawi* (picture wing group), *D. funebris*, *D. immigrans* or for any species of the *guaramunu*, *guarani*, *obscura*, and *tripunctata* groups. So, interesting intra-group polymorphisms in the status of presence/absence of Micropia sequences were solely identified for the *cardini* and *immigrans* groups. Fig 1 shows the species tree informing the presence and absence of Micropia related sequences in the genome of each of these species.

### Phylogenetic analysis, Micropia diversity, and potential coding sequences

As several intraspecific sequences clustered together in the BA phylogenetic tree obtained for the whole set of Micropia sequences recovered after the first filtering step (S3 Fig), the alignment could be reduced from 248 (S3 File) to 151 sequences (S4 File). The final Micropia phylogenetic tree reinforced reciprocal monophyly of several sets of sequences and confirmed the identity of the retrieved sequences, which were clustered with Micropia sequences obtained by [15] (Fig 2). Further evaluation of the recovered tree topology reveals the presence of four main clusters, which are listed here in ascending order of divergence into the tree: the first,

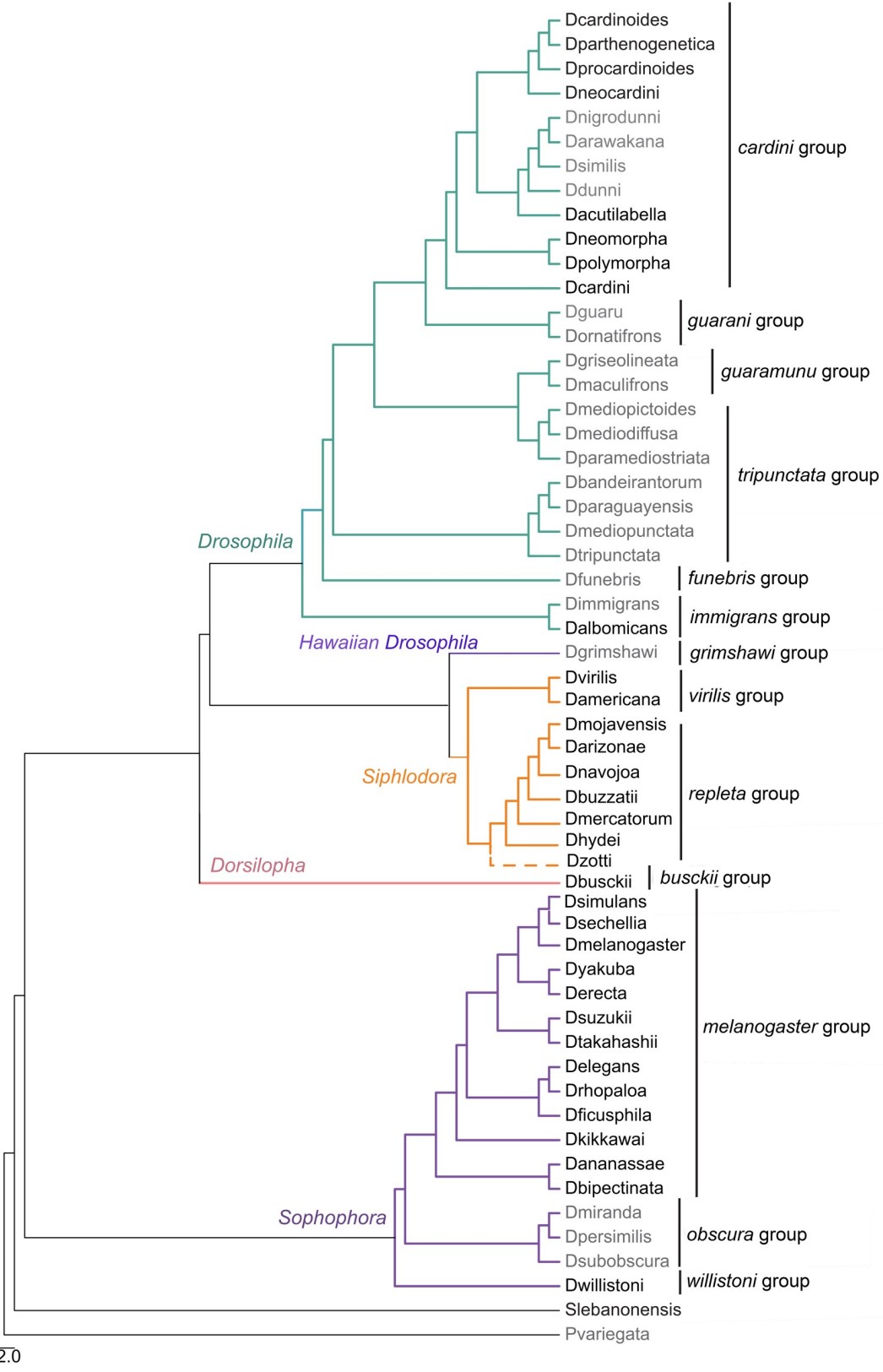

**Fig 1. Phylogenetic reconstruction of species analyzed in this study.** Phylogenetic reconstruction was based on data compiled from previous studies [50, 51, 52, 53, 54, 55 and 56] which have a limited overlap on sampled species. Species name in black represents the

presence of Micropia sequences and species name in grey represents the absence of such sequences. Distinct branch colors represent distinct subgenera within the *Drosophila* genus, and the classification follows [43]. *Drosophila* genus group species are also indicated to the right. *Scaptodrosophila* and *Phortica* are represented as outgroups of the *Drosophila* genus. The dashed line represents the potential phylogenetic position of *D. zottii*, since there is no molecular phylogeny neither any nuclear or mitochondrial gene available for this species.

presenting the Sacco sequences obtained by [15]; the second, grouping representatives of the Blastopia and MDG3 sequences obtained by [15]; the third, presenting the Bicca element recovered by [15]; and the fourth recovering all the Micropia sequences in a major polytomic clade, including sequences obtained by [15].

Following the criteria established by Capy et al. [30], we were able to recover 20 potential Micropia subfamilies based on monophyletic sequences (Fig 2) showing amino acid genetic divergences lower than 0.3 (Table 2 and S4 Table). Of these, nine subfamilies are monotypic and represented by a single sequence (subfamilies 2, 6, 8, 13, 16, 17, 18, 19 and 20). To the exception of subfamilies 4 and 15 (which were encountered only in species of the *melanogaster* group), all the remaining Micropia subfamilies are composed of species of distinct *Drosophila* species groups and subgenera.

As a result, there are clear cases of incongruence between the species and TE's phylogenies (Figs 1 and 2, respectively), in which Micropia sequences found in the genomes of distantly related species are clustered in the same subfamily in the Micropia phylogeny, and copies within a unique genome do not share a unique and exclusive common ancestor. For example, subfamily 7 (Fig 2) comprises sequences within the genome of *cardini* and *repleta* group species, belonging to the *Drosophila* and *Siphlodora* subgenera, respectively, together with sequences encountered within the genome of *D. willistoni*, which belongs to the *Sophophora* subgenus. As concerns the presence of divergent copies within the same genome, the cases of *D. buzzatii* (*repleta* group), *D. americana* (*virilis* group) and *D. willistoni* (*willistoni* group) should be highlighted, since the Micropia sequences present in the genomes of these species are widely spread over the tree, nested in five, six and nine of the subfamilies, respectively.

The analysis of potential coding sequences for the Micropia elements shown in the final tree (sequences of [15] were not included in this analysis, as well as the outgroup Copia-like sequence) shows that approximately 48% of them (48 from 100) putatively encode for reverse transcriptase enzyme (S5 Table). In fact, from the total set of 34 species with Micropia sequences evaluated here, only *D. erecta*, *D. kikkaway*, *D. mojavensis*, and *D. polymorpha* do not possess potentially encoding sequences.

## dS estimates and identification of horizontal transposon transfer (HTT) events

The use of *Adh*, *Amd* and *Ddc* nuclear gene sequences held a total of 4,367, 4,370 and 4,558 pairwise dS comparisons, respectively (S6 Table). Micropia dS values were lower than those found for the host nuclear genes in 277 cases (significance at the Fisher's exact test—with p-value $< 0.05$—were obtained for 96, 266 and 207 comparisons involving *Adh*, *Amd* and *Ddc*, respectively), revealing incompatible patterns with vertical transposon transmission (VTT). Thus, signals of HTTs account for 2.2%, 6.1% and 4.5% of the comparisons performed with *Adh*, *Amd* and *Ddc*, respectively. Fig 2 highlights all species involved in at least one case of significantly lower Micropia dS value. Indeed, only 19 of 97 sequences of Micropia for which the Fisher's Exact Test could be performed do not present any signal of involvement in HTTs events (sequences of [15] were not included in this analysis, as well as that from the outgroup and from *D. zotti*, for which none of the three nuclear genes have been previously

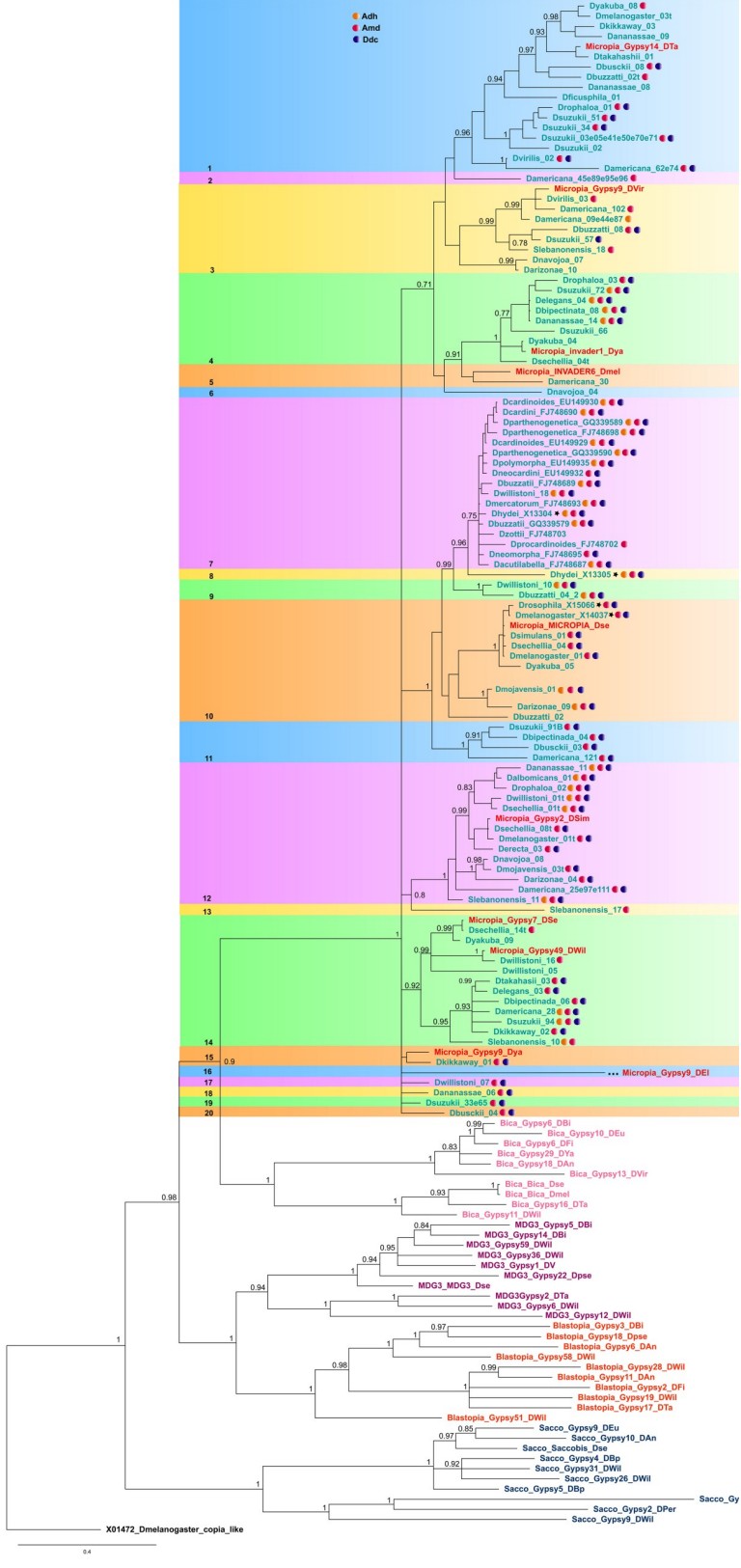

**Fig 2. Bayesian phylogenetic tree of the Drosophilidae Micropia sequences analyzed in this study after the second filtering step.** The phylogenetic tree was based on amino acid sequences following a mixed evolution model with

gamma correction. Bargues and Lerat´s sequences [15] were included in the analysis. Numbers from 1 to 20 on the left represent the Micropia subfamilies recovered in our data. Filled circles after Micropia sequence names indicate sequences involved in possible HTT events based on one-tailed Fisher's exact test involving pairwise comparisons of dS values between Micropia and nuclear genes (*Adh* in orange, *Amd* in pink, *Ddc* in purple; see S6 Table). Stars represent the four best-characterized Micropia elements (*D. hydei* dhMiF2 and dhMiF8; and *D. melanogaster* Dm11 and Dm2). The posterior probability of each clade is indicated beside its respective internal branch.

characterized). Concerning divergence times, most sequences presenting signals of HTT seem to have diverged during the last 20 mya (S6 Table).

## Discussion

### Micropia classification

By comparing our data with those of Bargues and Lerat's [15], it is possible to show that our non-stringent methodology retrieved sequences belonging to Micropia within the Micropia/Sacco group of the *Ty3/gypsy* retrotransposable elements. Within this group, Micropia is recovered as a monophyletic lineage and sister to the Bica group of LTR retroelements. The Bayesian phylogeny of these sequences highlights the existence of a high array of divergent sequences, which are compatible with the subdivision of Micropia into specific groups. Nevertheless, the taxonomic status represented by these remains a matter of debate.

In fact, except for the very well accepted criteria used to classify TEs in classes and sub-classes proposed by [60], in general, there is no consensus over the criteria adopted to achieve TEs families and subfamilies [61]. Several authors used different strategies to identify new TE families and subfamilies, whether based on nucleotide and/or amino acid sequence similarities [30, 48, 62, 63, 64, 65, 66]. Given the abundance and diversity of TEs, a classification for eukaryotic TEs based uniquely on nucleotide similarities was proposed [48]. Nevertheless, given the absence of evolutionary criteria based on reciprocal monophyly, this system is yet

**Table 2. Mean pairwise amino acid genetic distances between Micropia subfamilies.**

| | 01 | 02 | 03 | 04 | 05 | 06 | 07 | 08 | 09 | 10 | 11 | 12 | 13 | 14 | 15 | 16 | 17 | 18 | 19 |
|---|---|---|---|---|---|---|---|---|---|---|---|---|---|---|---|---|---|---|---|
| Subf. 02 | 0.357 | | | | | | | | | | | | | | | | | | |
| Subf. 03 | 0.313 | 0.345 | | | | | | | | | | | | | | | | | |
| Subf. 04 | 0.327 | 0.343 | 0.297 | | | | | | | | | | | | | | | | |
| Subf. 05 | 0.385 | 0.407 | 0.338 | 0.335 | | | | | | | | | | | | | | | |
| Subf. 06 | 0.436 | 0.397 | 0.412 | 0.423 | 0.315 | | | | | | | | | | | | | | |
| Subf. 07 | 0.410 | 0.442 | 0.324 | 0.341 | 0.376 | 0.376 | | | | | | | | | | | | | |
| Subf. 08 | 0.690 | 0.739 | 0.533 | 0.620 | 0.625 | 0.496 | 0.302 | | | | | | | | | | | | |
| Subf. 09 | 0.407 | 0.521 | 0.328 | 0.360 | 0.368 | 0.351 | 0.181 | 0.433 | | | | | | | | | | | |
| Subf. 10 | 0.421 | 0.407 | 0.341 | 0.345 | 0.381 | 0.400 | 0.217 | 0.451 | 0.237 | | | | | | | | | | |
| Subf. 11 | 0.468 | 0.461 | 0.431 | 0.405 | 0.454 | 0.422 | 0.274 | 0.512 | 0.284 | 0.271 | | | | | | | | | |
| Subf. 12 | 0.435 | 0.425 | 0.360 | 0.368 | 0.480 | 0.441 | 0.366 | 0.635 | 0.384 | 0.361 | 0.426 | | | | | | | | |
| Subf. 13 | 0.664 | 0.401 | 0.580 | 0.574 | 0.549 | 0.751 | 0.518 | 0.586 | 0.584 | 0.535 | 0.573 | 0.437 | | | | | | | |
| Subf. 14 | 0.413 | 0.441 | 0.358 | 0.382 | 0.393 | 0.374 | 0.332 | 0.577 | 0.334 | 0.317 | 0.385 | 0.354 | 0.539 | | | | | | |
| Subf. 15 | 0.381 | 0.512 | 0.346 | 0.363 | 0.398 | 0.406 | 0.323 | 0.586 | 0.318 | 0.324 | 0.385 | 0.302 | 0.440 | 0.304 | | | | | |
| Subf. 16 | 1.499 | 0.857 | 1.345 | 1.425 | 1.338 | 1.460 | 1.437 | 1.368 | 1.401 | 1.359 | 1.302 | 1.484 | 1.408 | 1.391 | 1.507 | | | | |
| Subf. 17 | 0.406 | 0.511 | 0.349 | 0.375 | 0.418 | 0.426 | 0.297 | 0.586 | 0.313 | 0.327 | 0.398 | 0.292 | 0.431 | 0.336 | 0.140 | 1.365 | | | |
| Subf. 18 | 0.401 | 0.449 | 0.330 | 0.375 | 0.418 | 0.390 | 0.339 | 0.611 | 0.312 | 0.318 | 0.391 | 0.288 | 0.430 | 0.310 | 0.149 | 1.533 | 0.137 | | |
| Subf. 19 | 0.376 | 0.505 | 0.334 | 0.351 | 0.371 | 0.390 | 0.266 | 0.598 | 0.285 | 0.287 | 0.337 | 0.301 | 0.465 | 0.309 | 0.119 | 1.375 | 0.157 | 0.160 | |
| Subf. 20 | 0.366 | 0.460 | 0.330 | 0.356 | 0.309 | 0.432 | 0.233 | 0.496 | 0.238 | 0.285 | 0.367 | 0.247 | 0.651 | 0.280 | 0.145 | 1.509 | 0.138 | 0.123 | 0.113 |

widely controversial. So, we adopted here more conservative criteria, according to which different subfamilies are established based on reciprocal monophyly and divergence values higher than 0.3 at the amino acid level [30].

Adopting these criteria, our data shows the existence of at least 20 potential Micropia subfamilies that form the reciprocally monophyletic groups or monotypic lineages shown in Fig 2. Several of these subfamilies are spread over distinct *Drosophila* subgenera and species groups, although only subfamilies 7 and 12 could be sampled across species of *Sophophora*, *Drosophila*, and *Siphlodora*. In this sense, most sequences within the *Drosophila* subgenus species are clustered in subfamily 7, whereas sequences of *Siphlodora* are highly intermingled in the topology, but are predominantly nested in subfamilies 3, 7, 10 and 12. The other Micropia subfamilies are mostly comprised of sequences within species of the *Sophophora* subgenus, especially by sequences within the *melanogaster* group. Interestingly, sequences of Micropia used by [15] are distributed across nine of the 20 subfamilies here established, showing the wide diversity of Micropia sequences in Drosophilidae species genomes.

### Micropia evolutionary history

In addition to this pattern of high diversity, our data also show that the evolutionary history of Micropia retroelement in *Drosophila* is characterized by several VTTs and HTTs events. Although VTTs may comprise the predominant form of transmission (94–98% of the events), HTTs is clearly an important way that these genomic parasites have to evade genomic extinction [17, 18]. In our data, the evidence for HTT in Micropia evolution came from three main sources: (i) the patchy distribution within Drosophilidae phylogeny, (ii) the incongruence between Micropia and species phylogenies, and (iii) the significantly lower dS values presented by some Micropia sequences in comparison to nuclear host genes [17, 26]. In the first line of evidence, PCR and Dot-Blot analyses provided some interesting results, especially when they were evaluated considering the results obtained through genomic data, aiming to get inferences about presence/absence patterns along the Drosophilidae phylogeny. Sequence analysis was further performed using amino acid data to reconstruct the Micropia phylogenetic relationships and using codon-aligned nucleotide data in order to measure synonymous distances. This whole set of results enable to envision a hypothesis about the evolution of Micropia sequences within Drosophilidae.

The *cardini* group species was the best-represented *Drosophila* group in our analysis, and 80% of its species had their genome analyzed (12 from the 15 described species; [67]). Of these, eight species presented Micropia sequences. Conversely, the *melanogaster* and the *repleta* groups, for which several species have sequenced genomes, presented the higher percentage of species containing Micropia copies (100%). The number of isolated sequences is generally higher for species belonging to these groups, for which whole genome sequences are frequently available. Nevertheless, the use of *in vitro* methodologies to investigate the presence of TEs in non-model group species revealed here an important strategy to establish a robust evolutionary hypothesis for the element. For example, using such methodologies we were able to identify the absence of Micropia copies in the genome of several species belonging to distinct groups (*funnebris*, *guaramunu*, *guarani*, *immigrans*, and *tripunctata*), confirming, therefore, the patchy distribution of Micropia in the *Drosophila* subgenus.

The *cardini* group species showed an interesting Micropia distribution pattern. Micropia sequences are present only in the genome of species occurring in the mainland, from south North America to southern South America [68]. The other four species, *D. arawakana*, *D. dunni*, *D. nigrodunni*, and *D. similis*, which seem to be devoid of Micropia (S1 Fig), are endemic to the Caribbean islands [68]. The clustering of the Micropia sequences presented by

the mainland *cardini* species and their straightforward similarity in amino acid sequences suggest that the element has invaded the genome of these species around 1.5 mya, which is much more recent than the divergence times estimated for the target species (4–35 mya, as estimated by [52]). Considering this, it is interesting to note that 73% (8 of 11) of the Micropia RT sequences analyzed for the *cardini* group species seem to be capable of coding for reverse transcriptase enzyme, which is also evidence in favor of a recent invasion. This invasion apparently occurred through multiple HTTs, as can be inferred through the comparison of pairwise Micropia dS values and orthologous nuclear genes dS values. This methodology is able to detect HTTs between closely-related species [29]. In fact, all the 51 comparisons involving only species of the *cardini* group showed significantly lower dS values for Micropia than for any of the three evaluated nuclear genes. Nevertheless, although several HTTs events seem to have occurred between species of the *cardini* group, it is quite probable that the ancestor sequence of this group came from a species belonging to the *repleta* group (or another related group not analyzed here), for which at least some sequences from subfamily 7 seem to have evolved through VTTs. This can be seen, for example, by the absence of rejection of the null hypothesis of VTT in the comparison of dS values between the sequences Dhydei_X13304 and Dbuzzatti_04_2 and those of the host nuclear genes. This pattern is also corroborated by [39].

Several other HTTs might also have occurred within the *melanogaster* group (53.3% of potential coding sequences) and evidence for these can be found within subfamilies 1, 4, 10, 11 and 14. In subfamily 10, for example, the Micropia copies in *D. melanogaster*, *D. simulans* and *D. sechellia* genomes are identical, suggesting recent events of HTTs. Conversely, in subfamily 1, there are clear incongruences between Micropia and species phylogeny, and a sequence encountered in *D. suzukii* may have been recently transferred to *D. rhopaloa*, given the earlier branching of the Micropia sequences from *D. suzukii* genome. This event occurred around 5 mya. In fact, these species are included in different subgroups of the *melanogaster* group, for which divergence times at the same divergence level are older than 10 mya [46].

Interestingly, signals of HTTs are less straightforward among species of the *repleta* group, and despite the presence of sequences nested in different Micropia subfamilies; only subfamily 7 presents some evidence of HTT involving *D. hydei*, *D. buzzatii* and *D. mercatorum*. Such events were dated to approximately 1.25 mya, which is quite more recent than the divergence times estimated for these species (4–16 mya [52]). There are two common features between these events and those presented above for the *cardini* group: also here multiple HTTs can be inferred, and these lie in the same confidence interval time as those discussed above. Moreover, all the evaluated species of both the *cardini* and the *repleta* groups occur in the Neotropics [67], which faced severe climatic oscillations during this period [69]. Since it was already shown that these events possibly changed the distribution of several species of *Drosophila* [70, 71], they may have led to several secondary contacts which created the necessary conditions for HTT.

All the HTTs discussed so far occurred between closely related species, comprising the same species group. According to [16], it is expected that the more species sampled within a group, the more HTT events will be discovered, since retrotransposons show low HTT rates between distantly related lineages. Nevertheless, considering the dS comparisons performed within each of the Micropia subfamilies, in association to the incongruences between species and Micropia phylogenies, we were also able to hypothesize the occurrence at least seven other HTTs involving species from distinct *Drosophila* groups or even distinct subgenera, as follow:

- Subfamily 3: since this subfamily is widely spread in the genome of species belonging to the subgenus *Siphlodora*, there must have occurred one HTT from one species of the *Siphlodora*

subgenus to *D. suzukii*, the only species of the *melanogaster* group with sequences belonging to this Micropia subfamily;

- Subfamily 7: the sequences Dhydei_X13304 and X13305 do not present signals of HTT with Dbuzzatti_04_2, so these sequences might be the presumably ancestral copies within this subfamily. In this way, besides the HTTs within the *cardini* and *repleta* groups discussed above, and that from one species of the *repleta* group (possibly *D. hydei*) to another species of the *cardini* group, there might have occurred at least one HTT from *D. buzzatii* to *D. willistoni*;

- Subfamily 11: as Damericana_121 does not show signals of HTT comparing with Dbusckii_03, they might represent ancestral sequences. In this way, it might have occurred at least one HTT to species of the *melanogaster* group;

- Subfamily 12: given the absence of HTTs signals among several species of the *melanogaster* group, as well as among species of the *Siphlodora* subgenus, most of these copies possibly evolved through VTT since the most recent common ancestor (MRCA) of both lineages. Nevertheless, there is evidence of one HTT presumably from *D. sechellia* to *D. willistoni*, one from *D. ananassae* to *D. albomicans*, and one involving the MRCA of the *melanogaster* and *Siphlodora* lineages.

- Subfamily 14: this Micropia subfamily is widespread in the *melanogaster* group, from which an HTT presumably occurred to *D. americana*.

In conclusion, the Micropia evolutionary history is based on VTTs and HTTs events with a high diversification of sequences leading to the distinct subfamilies here detected, with some sequences still capable to encode RT enzyme. Moreover, species from the *repleta* and *melanogaster* group seem to have played an important role in most HTT events inferred here within *Drosophila*. The wide distribution range occupied by some species of these groups possibly contributed to these phenomena, by providing more chances to HTT due to ancient overlapping distribution with other species [16].

## Supporting information

**S1 Fig. In vitro searches for Micropia within genomes.** A: PCR-blot results of species from the *cardini* and *repleta* groups. B: Dot-blot on genomic DNA confirming the pattern seen on the PCR-blot. In both cases, the probe used was an 812bp PCR fragment from *D. hydei* dhMiF2 sequence. Control: 5μl (in 10 μl) of the Micropia probe.
(TIF)

**S2 Fig. In vitro searches for Micropia within genomes.** Dot-blot on genomic DNA. The probe used was an 812bp PCR fragment from *D. hydei* dhMiF2 sequence. 1. *D. funnebris*; 2. *D. griseolineata*; 3. *D. maculifrons*; 4. *D. guaru*; 5. *D. ornatifons*; 6. *D. immigrans*; 7. *D. bandeirantorum*; 8. *D. mediodiffusa*; 9. *D. mediopictoides*; 10. *D. mediopunctata*; 11. *D. paraguayensis*; 12. *D. paramediostriata*; 13. *D. tripunctata*. +: positive control, 5μl (in 10 μl) of Micropia probe; -: negative control, *D. similis* DNA.
(TIF)

**S3 Fig. Bayesian phylogenetic tree of the 247 Micropia sequences recovered by our searches within the Drosophilidae species analyzed in this study after the first filtering strategy.** The phylogenetic tree was based on amino acid sequences following a mixed evolution model with gamma correction. Bargues and Lerats´ sequences [15] were included in the analysis. The

posterior probability of each clade is indicated beside its respective internal branch.
(TIF)

**S1 Table. List of BLASTn and tBLASTn results.** Species scaffold: represents the scaffold in the species genome where the Micropia sequence was found. First nt: first nucleotide in the scaffold where the Micropia RT sequence homologous to our query was detected. Last nt: last nucleotide in the scaffold where the Micropia RT sequence homologous to our query was detected. BLAST id: Blast identities. E-value: E-value recovered by BLAST searches. Methodology: database and *in silico* search methodology used to find the Micropia best match query. *Sequences used as initial BLAST searches. **Sequences remained after the two BLAST search methodology.
(XLSX)

**S2 Table. GenBank accession numbers of nuclear genes used in the dS analysis.** Data not available.
(XLSX)

**S3 Table. Summary of the number of Micropia sequences recovered in each BLAST search step.** Here includes the sequences obtained within the Drosophilidae genomes and the sequences used as query (*).
(XLSX)

**S4 Table. Amino acid genetic distances between sequences belonging to the same Micropia subfamily.** Data for each subfamily are in distinct sheets in this Excel file.
(XLSX)

**S5 Table. Potentially coding sequences and their respective coding frame.** Sequences presenting stop codons are represented by a dash (-). The involvement in HTT was identified by the Fisher's exact test (see S6 Table)
(XLSX)

**S6 Table. Pairwise comparative analysis of dS values between Micropia and Adh, Amd and Ddc nuclear gene sequences.** Comparisons suggesting horizontal transposon transfer events were statistically tested by one-sided tail Fisher's exact test (Ost). Colors represent the p values lower than 0.05 (see Fig 2) to $Ost_{Micropia-Adh}$ (orange), $Ost_{Micropia-Amd}$ (pink) and $Ost_{Micropia-Ddc}$ (purple).
(XLSX)

**S1 File. Nucleotide alignment comprising all Micropia sequences retrieved.** The 363 sequences were recovered through *in vitro* and *in silico* searches. The sequence used as outgroup (a Copia retroelement sequence from *D. melanogaster* genome) were added to this alignment.
(FAS)

**S2 File. Alignment of nucleotide sequences.** The sequences from S1 File were filtered to include only the ones showing a minimum overlap of 300 bp (first filtering strategy) encompassing 247 Micropia sequences. The sequence used as outgroup (a Copia retroelement sequence from *D. melanogaster* genome) were added to this alignment.
(FAS)

**S3 File. Alignment of amino acid sequences.** This alignment comprises the 247 amino acid Micropia sequences plus the sequence used as outgroup (a Copia retroelement sequence from *D. melanogaster* genome) recovered after the first filtering strategy and employed for the

assessment of reciprocal monophyly patterns regarding sequences retrieved from the same species (see S3 Fig).
(FAS)

**S4 File. Alignment of amino acid sequences.** This alignment comprises the 100 amino acid Micropia sequences recovered after the second filtering strategy, plus the 50 sequences of the Micropia/Sacco group characterized by [15], also including the sequence used as outgroup (a Copia retroelement sequence from *D. melanogaster* genome) employed in the phylogenetic reconstruction of Fig 2.
(FAS)

**S5 File. Codon alignment of nucleotide sequences.** This alignment comprises the 100 nucleotide Micropia sequences recovered after the second filtering strategy employed in the dS estimates.
(FAS)

## Acknowledgments

We thank Dr. Emmanuele Lerat for kindly providing the sequences belonging to Micropia/ Sacco group of Ty3/Gypsy retroelements, Tiago Ribeiro for suggestions on the early version of the manuscript; the two anonymous reviewers for valuable comments to improve the manuscript; and all researchers from the Laboratório de Drosophilidae at UFRGS.

## Author Contributions

**Conceptualization:** Juliana Cordeiro, Vera Lúcia da Silva Valente.

**Investigation:** Juliana Cordeiro, Tuane Letícia Carvalho.

**Methodology:** Juliana Cordeiro, Tuane Letícia Carvalho, Lizandra Jaqueline Robe.

**Supervision:** Vera Lúcia da Silva Valente, Lizandra Jaqueline Robe.

**Writing – original draft:** Juliana Cordeiro, Tuane Letícia Carvalho, Lizandra Jaqueline Robe.

**Writing – review & editing:** Juliana Cordeiro, Tuane Letícia Carvalho, Lizandra Jaqueline Robe.

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
