## [Decision Letter · Decision Letter 0]

19 Aug 2019

PONE-D-19-19755

Evolutionary history and classification of Micropia retroelements in Drosophilidae species

PLOS ONE

Dear Dr Cordeiro,

Thank you for submitting your manuscript to PLOS ONE. After careful consideration, we feel that it has merit but does not fully meet PLOS ONE’s publication criteria as it currently stands. Therefore, we invite you to submit a revised version of the manuscript that addresses the points raised during the review process.

We would appreciate receiving your revised manuscript by Oct 03 2019 11:59PM. To enhance the reproducibility of your results, we recommend that if applicable you deposit your laboratory protocols in protocols.io, where a protocol can be assigned its own identifier (DOI) such that it can be cited independently in the future. For instructions see: http://journals.plos.org/plosone/s/submission-guidelines#loc-laboratory-protocols

We look forward to receiving your revised manuscript.

Kind regards,

Ruslan Kalendar, PhD

Academic Editor

PLOS ONE

Journal Requirements:

Reviewers' comments:

Reviewer's Responses to Questions

**Comments to the Author**

1. Is the manuscript technically sound, and do the data support the conclusions?

Reviewer #1: Yes

Reviewer #2: Partly

2. Has the statistical analysis been performed appropriately and rigorously? 

Reviewer #1: Yes

Reviewer #2: Yes

3. Have the authors made all data underlying the findings in their manuscript fully available?

Reviewer #1: No

Reviewer #2: No

4. Is the manuscript presented in an intelligible fashion and written in standard English?

Reviewer #1: No

Reviewer #2: Yes

5. Review Comments to the Author

Reviewer #1: The authors of this study have previously published articles regarding transposable elements in the Drosophilidae. This study includes a large taxon sampling. Two techniques were used to sample species for presence or remnants of the Micropia TE. One technique was to sample “in vitro” from fly DNA isolated by the authors. The other sampling technique to search through published genomes referred to as “in silico.” The evolution of Micropia elements uses previously established and employed criteria of percent similarity.

Overall:

Slight editing for word usage and grammar needed. For example

Line 28 “detaches” is incorrect usage. I would omit “detaches as” and replace with “is”

Line 38 “identified combined” omit “combined”

Line 41 “sequences found in” omit “found”

Line 50 “McClintock” add ‘s to be “McClintock’s”

With regard to the taxon sampling:

For the natural populations for 24 Drosophila species - were these field collected by the authors? Have any morphological vouchers been deposited in a collection?

Line 132 “Genomic DNA was prepared according to [44].” This paper should briefly summarize ref 44’s DNA prep procedure rather than expecting the reader to chase down publications to evaluate a study. Was the isolation through single fly preps or using a large quantity of flies such as 2 ml volume ground using a grinder or container that is reused. Previous horizontal transfer has been misidentified when a grinder was reused due to the sensitivity of PCR amplification.

It was well-thought-out to use of three nuclear genes for comparison for rates of change in sequence divergence in the Micropia TE.

Lines 261 and 262 The phylogenetic tree of the species used in this study “…was based on was based on data compiled from [49, 50, 51, 52, 53, 54 and 55].” Was this tree created by stitching together clades from these papers because there is no one taxonomic investigation that overlaps the species in this investigation? If this is correct this should be clearly stated to the reader.

Reviewer #2: Cordeiro et al studied the phylogenetic dristribution of micropia sequences and showed that HTT can be an important component of the evolution history of micropia. The manuscript is well structured and the logic is sound. However, the method, especially for the in silico part, is too loose and may bias the results. The manuscript is also vague in methods and missing some important information (dS, divergence times, alignment, etc.) In addition, there are many grammar mistakes and writing need to be improved. Therefore, I recommend a major revision.

Below are some more specific comments:

The matching threshold of In silico searches were not stringent enough and there could be some false positives. I thus recommend the authors blasting with lower e-value threshold.

The authors should also provide more details of the In silico search process, e.g. what database was used for blast.

It would be better if the authors provide more statistics for the In silico searchs, e.g. how many hits were kept during each Blastn/tBlastn process.

Line 50: This sentence read awkwardly.

Line 60: The authors need to clarify what LTR stands for.

Line 71: grammar mistakes and typo

Line 170: The default e-value is usually too high.

Line 177: Scores are dependent on gene length. I recommend to report e-value instead.

Line 180: This sentence in confusing. Not sure what does it mean.

Line 189: The authors need to explain why doing another round of Blast.

Line 194: Why translate unaligned sequences? It seems to me that unaligned sequences were not micropia based on sequence similarity.

Line 193: The authors need to provide the alignment sequences as a figure/table.

Line 200: Manually removing gaps may bias the phylogenetic analyses

Line 206: The authors need to justify why D.melanogaster was used as a outgroup.

Line 225: It would be better if the authors could provide the dS and divergence times as a table

6. PLOS authors have the option to publish the peer review history of their article (what does this mean?). If published, this will include your full peer review and any attached files.

Reviewer #1: No

Reviewer #2: No

---

## [Author Response · Author response to Decision Letter 0]

3 Oct 2019

Dear Reviewers,

Please find attached the MS entitled “Evolutionary history and classification of Micropia retroelements in Drosophilidae species” (PONE-D-19-19755) which is a thoroughly revised version of the previous submission to PLOS ONE. We thank you for the opportunity as well as for the constructive criticism from both reviewers which allowed major improvements in our study.

We followed carefully all recommendations and provide below a point-by-point explanation on how we proceeded in regard to each specific comment of the reviewers. We are confident the current version is much improved and hope it reaches the level of quality expected by PLOS ONE.

We tracked the changes suggested by the reviewer by marking them in light blue in the text, and we believe that now all data underlying the findings are fully available as supplementary material. 

Sincerely,

J. Cordeiro, on behalf of all authors

Reviewer #1: The authors of this study have previously published articles regarding transposable elements in the Drosophilidae. This study includes a large taxon sampling. Two techniques were used to sample species for presence or remnants of the Micropia TE. One technique was to sample “in vitro” from fly DNA isolated by the authors. The other sampling technique to search through published genomes referred to as “in silico.” The evolution of Micropia elements uses previously established and employed criteria of percent similarity.

Reply: Thank you for all your constructive comments and for the opportunity to improve our manuscript. All requests and suggestions were addressed and we are confident that we now provided a more comprehensive manuscript.

Overall:

Slight editing for word usage and grammar needed. For example

Line 28 “detaches” is incorrect usage. I would omit “detaches as” and replace with “is”

Line 38 “identified combined” omit “combined”

Line 41 “sequences found in” omit “found”

Line 50 “McClintock” add ‘s to be “McClintock’s”

Reply: We revised the MS for grammar and professional English use and the suggestions provided by the reviewer were implemented.

With regard to the taxon sampling:

For the natural populations for 24 Drosophila species - were these field-collected by the authors? Have any morphological vouchers been deposited in a collection?

Reply: The species were kept in the Laboratory of Drosophilidae at Universidade Federal do Rio Grande do Sul, Brazil. Some species were obtained through the previous Tucson Drosophila Stock Center (current The National Drosophila Species Stock Center at Cornell University). Some of them were field collected by the researchers Dr. Marco Silva Gottschalk, Dr. Daniela De Toni, Dr. Jonas Döge and Dr. Luciano Basso da Silva during 2000-2009. In these cases, species vouchers are available at the Laboratory of Drosophilidae. In this way, we identified all collectors/suppliers in Table 1 and provided the above information in the main text.

Line 132 “Genomic DNA was prepared according to [44].” This paper should briefly summarize ref 44’s DNA prep procedure rather than expecting the reader to chase down publications to evaluate a study. Was the isolation through single fly preps or using a large quantity of flies such as 2 ml volume ground using a grinder or container that is reused. Previous horizontal transfer has been misidentified when a grinder was reused due to the sensitivity of PCR amplification.

Reply : In the MS, we improved this sentence to “Genomic DNA was extracted through phenol-isoamyl-chloroform protocol according to [44] with approximately 100 adult flies per species macerated in liquid nitrogen using individual new sterile grinders.”

Lines 261 and 262 The phylogenetic tree of the species used in this study “…was based on was based on data compiled from [49, 50, 51, 52, 53, 54 and 55].” Was this tree created by stitching together clades from these papers because there is no one taxonomic investigation that overlaps the species in this investigation? If this is correct this should be clearly stated to the reader.

Reply : Thanks for calling attention to this issue. Yes, there is only limited species overlap in the available phylogenetic studies so far. We now clearly stated this in the main text and in the legend of Figure 1.

Reviewer #2: Cordeiro et al studied the phylogenetic dristribution of micropia sequences and showed that HTT can be an important component of the evolution history of micropia. The manuscript is well structured and the logic is sound. However, the method, especially for the in silico part, is too loose and may bias the results. The manuscript is also vague in methods and missing some important information (dS, divergence times, alignment, etc.) In addition, there are many grammar mistakes and writing need to be improved. Therefore, I recommend a major revision.

Reply: We want also to thank you for your constructive comments. They were essential to improve our manuscript. All requests, questions, and suggestions were addressed and we are confident that we now provided a more comprehensive manuscript.

Below are some more specific comments:

The matching threshold of In silico searches were not stringent enough and there could be some false positives. I thus recommend the authors blasting with lower e-value threshold.

The authors should also provide more details of the In silico search process, e.g. what database was used for blast. 

It would be better if the authors provide more statistics for the In silico searchs, e.g. how many hits were kept during each Blastn/tBlastn process.

Reply: Thanks for the comments. The Methodology section was improved to address the questions regarding E-value and target databases. Furthermore, basic statistics of the results were provided at the beginning of the Results section.

We emphasize that the adopted E-value needed to be less stringent in order to recover divergent sequences and allow a confident description of the Micropia diversity and subdivision. Otherwise, it wouldn’t be possible to suggest a classification scheme for this retroelement. The effectiveness of the adopted strategy can be further accessed by the recovered phylogenetic tree, which supports the reciprocal monophyly of all retrieved Micropia sequences.

Line 50: This sentence read awkwardly.

Reply: The sentence was improved.

Line 60: The authors need to clarify what LTR stands for.

Reply: The acronym LTR was clarified through the use of long terminal repeats in the first appearance of LTR.

Line 71: grammar mistakes and typo

Reply: We improved the MS with English revision also correcting typos.

Line 170: The default e-value is usually too high.

Reply: We did not use a default E-value. A threshold of 1.0E-05 was adopted in our searches together with a minimum score of 50, and this is now clarified in the text.

Line 177: Scores are dependent on gene length. I recommend to report e-value instead.

Reply: Since both, scores and e-values, present some shortcomings, we are presenting both values now.

Line 180: This sentence is confusing. Not sure what does it mean.

Reply: The sentence was improved to “Sequences that failed to align in this first multiple alignment steps underwent a second alignment step (this time pairwise or even local alignment) against the query sequence that presented the highest score in the BLASTn searches (hereafter “best query” sequence). This is certainly an unusual strategy, which was necessary in order to align some divergent sequences.

Line 189: The authors need to explain why doing another round of Blast.

Reply: We performed this two BLAST step strategy aiming to achieve a better representation of the whole diversity of Micropia sequences encountered in Drosophilidae, in order to attain our goal of providing a classification scheme for this retroelement. In this line, we added the following sentence to the manuscript: “This two BLAST step strategy was performed to guarantee that the real diversity of Micropia sequences was retrieved from the genomes, enabling a better representation of these sequences in our dataset.”

Line 194: Why translate unaligned sequences? It seems to me that unaligned sequences were not micropia based on sequence similarity.

Reply: Translation in all reading frames was performed with unaligned sequences in order to identify putatively encoding elements. If aligned sequences were used in this step, encoding sequences that did not present insertions encountered in other sequences could have been erroneously classified as inactive. The effectiveness of the adopted strategy can be further accessed by the recovered phylogenetic tree, which supports the reciprocal monophyly of all Micropia sequences.

Line 193: The authors need to provide the alignment sequences as a figure/table.

Reply: Thanks for calling attention to this issue. We now provide the nucleotide and amino acid alignments employed in each step of the methodology as .fas fasta files in the supplementary material (S1 – S5 Files).

Line 200: Manually removing gaps may bias the phylogenetic analyses

Reply: In TEs studies, this is a common strategy that allows amino acid and dS analysis of all obtained sequences [see, for example, Ludwig & Loreto (2007) and Mota et al. (2010)]. As TEs usually present frameshifts, if such a strategy was not adopted, only nucleotide sequences or amino acid sequences of potentially encoding Micropia sequences could be analyzed. The first of these possibilities does not usually attain a good phylogenetic resolution, whereas the second only provides a partial description of the evolutionary scenario. Thus, only the manual edition of gaps that allows leaving all sequences in frame turned it possible to reach our aim of providing a classification scheme for Micropia sequences (by providing a resolved phylogenetic tree) and to infer putative HTTs (by enabling comparisons of dS estimates).

References:

Ludwig, A., & Loreto, E. L. S. (2007). Evolutionary pattern of the gtwin retrotransposon in the Drosophila melanogaster subgroup. Genetica, 130(2), 161-168.

Mota, N. R., Ludwig, A., da Silva Valente, V. L., & Loreto, E. L. S. (2010). Harrow: new Drosophila hAT transposons involved in horizontal transfer. Insect Molecular Biology, 19(2), 217-228.

Line 206: The authors need to justify why D.melanogaster was used as a outgroup.

Reply: In this study, we used a Copia-like transposable element sequence as outgroup as it belongs to a distinct transposable element superfamily (Ty1/Copia) than Micropia (Ty3/Gypsy) (Bargues and Lerat 2017). This Copia-like retroelement was first found in the D. melanogaster genome (Saigo et al. 1984). We improved the sentence to better comprehension.

References: 

Bargues N, Lerat E. Evolutionary history of LTR retrotransposons among 20 Drosophila species Mobile DNA. 2017; 8:7

Saigo,K., Kugimiya,W., Matsuo,Y., Inouye,S., Yoshioka,K., Yuki, S. (1984) Identification of the coding sequence for a reverse transcriptase-like enzyme in a transposable genetic element in Drosophila melanogaster Nature 312: 659–661

Line 225: It would be better if the authors could provide the dS and divergence times as a table

Reply: We now provide the divergence time of Micropia sequences within supplementary S5 Table.

---

## [Editor Report · Decision Letter 1]

7 Oct 2019

Evolutionary history and classification of Micropia retroelements in Drosophilidae species

PONE-D-19-19755R1

Dear Dr. Cordeiro,

We are pleased to inform you that your manuscript has been judged scientifically suitable for publication and will be formally accepted for publication once it complies with all outstanding technical requirements.

With kind regards,

Ruslan Kalendar, PhD

Academic Editor

PLOS ONE

---

## [Editor Report · Acceptance letter]

10 Oct 2019

PONE-D-19-19755R1 

Evolutionary history and classification of Micropia retroelements in Drosophilidae species 

Dear Dr. Cordeiro:

I am pleased to inform you that your manuscript has been deemed suitable for publication in PLOS ONE. Congratulations! Your manuscript is now with our production department. 

With kind regards,

on behalf of

Dr. Ruslan Kalendar 

Academic Editor

PLOS ONE